# Assessment of utilization of automated systems and laboratory information management systems in clinical microbiology laboratories in Thailand

Preeyarach Klaytong[1], Panida Chamawan[2], Voranadda Srisuphan[2], Krittiya Tuamsuwan[2], Phairam Boonyarit[2], Adisak Sangchankoom[3], Archawin Rojanawiwat[4], Aekkawat Unahalekhaka[4], Kulsumpun Krobanan[4], Pimrata Leethongdee[4], Orapan Sripichai[4], Vanaporn Wuthiekanun[1], Paul Turner[5,6], Direk Limmathurotsakul[1,6,7]*

1 Mahidol Oxford Tropical Medicine Research Unit, Faculty of Tropical Medicine, Mahidol University, Bangkok, Thailand, 2 Health Administration Division, The Office of Permanent Secretary, Ministry of Public Health, Nonthaburi, Thailand, 3 Microbiology Laboratory, Udonthani Hospital, Udonthani, Thailand, 4 Department of Medical Science, Ministry of Public Health, Nonthaburi, Thailand, 5 Cambodia Oxford Medical Research Unit (COMRU), Angkor Hospital for Children, Siem Reap, Cambodia, 6 Nuffield Department of Medicine, Centre for Tropical Medicine and Global Health, University of Oxford, Oxford, United Kingdom, 7 Department of Tropical Hygiene, Faculty of Tropical Medicine, Mahidol University, Bangkok, Thailand

* direk@tropmedres.ac

## Abstract

### Introduction

Clinical microbiology laboratories are essential for diagnosing and monitoring antimicrobial resistance (AMR). Here, we assessed the systems involved in generating, managing and analyzing blood culture data in these laboratories in an upper-middle-income country.

### Methods

From October 2023 to February 2024, we conducted a survey on the utilization of automated systems and laboratory information management systems (LIMS) for blood culture specimens in 2022 across 127 clinical microbiology laboratories (one each from 127 public referral hospitals) in Thailand. We categorized automated systems for blood culture processing into three steps: incubation, bacterial identification, and antimicrobial susceptibility testing (AST).

### Results

Of the 81 laboratories that completed the questionnaires, the median hospital bed count was 450 (range, 150-1,387), and the median number of blood culture bottles processed was 17,351 (range, 2,900-80,330). All laboratories (100%) had an automated blood culture incubation system. Three-quarters of the laboratories (75%, n=61) had at least one automated system for both bacterial identification and AST, about a quarter (22%, n=18) had no automated systems for either step, and two laboratories (3%) outsourced both steps.

**Data availability statement:** The dataset generated for this study is available from the Figshare data repository (https://doi.org/10.6084/m9.figshare.28028285).

**Funding:** The study was supported by Ministry of Public Health Thailand and Wellcome Trust of Great Britain (224681/Z/21/Z). For the purpose of Open Access, the author has applied a CC BY public copyright license to any Author Accepted Manuscript version arising from this submission. The funders had no role in study design, data collection and analysis, decision to publish, or preparation of the manuscript.

**Competing interests:** The authors have declared that no competing interests exist.

The systems varied and were associated with the hospital level. Many laboratories utilized both automated systems and conventional methods for bacterial identification (n = 54) and AST (n = 61). For daily data management, 71 laboratories (88%) used commercial microbiology LIMS, three (4%) WHONET, three (4%) an in-house database software and four (5%) did not use any software. Many laboratories manually entered data of incubation (73%, n = 59), bacterial identification (27%, n = 22) and AST results (25%, n = 20) from their automated systems into their commercial microbiology LIMS. The most common barrier to data analysis was 'lack of time', followed by 'lack of staff with statistical skills' and 'difficulty in using analytical software'.

## Conclusion

In Thailand, various automated systems for blood culture and LIMS are utilized. However, barriers to data management and analysis are common. These challenges are likely present in other upper-middle-income countries. We propose that guidance and technical support for automated systems, LIMS and data analysis are needed.

## Introduction

Clinical microbiology laboratories are the cornerstone of diagnosing, managing and monitoring antimicrobial resistant (AMR) infection [1]. It is recommended that referral hospitals have clinical microbiology laboratories for bacterial culture, and antimicrobial susceptibility testing (AST) [2]. However, almost half of the world's population lacks access to basic diagnostics, and only 1.3% of the 50,000 clinical laboratories in sub-Saharan African conduct bacterial cultures [3]. In addition, laboratory data management in low and middle-income countries (LMICs) is under-resourced, limiting progress towards comprehensive AMR surveillance [4]. Clinical microbiology laboratories also require a fit-for-purpose microbiology laboratory information management system (LIMS) given the complexity of specimen management, testing methods and result data.

Thailand is an upper-middle-income country with a strong universal health coverage (UHC) and a robust health infrastructure at relatively low cost [5]. Barriers to blood culture sampling in Thailand are less common than in other LMICs [6]. In the internal medicine and pediatric departments, most patients (>80%) presenting with severe infection have a blood culture taken within ±1 calendar day of the start of parenteral antibiotic treatment [7]. Consequently, clinical microbiology laboratories in Thailand have a higher capacity compared to other LMICs even after adjusting for health expenditure [8]. Thailand also has a national laboratory-based AMR surveillance programme, reporting the percentage of susceptible organisms isolated from clinical specimens nationwide since 1998 [9]. Nonetheless, data from clinical microbiology laboratories alone cannot determine whether an AMR infection is of community-origin or hospital-origin, as defined by WHO GLASS [10], and cannot monitor the total number of deaths following AMR infections [11,12].

In 2020, we developed the AutoMated tool for Antimicrobial resistance Surveillance System (AMASS), an offline application that allows hospitals to automatically analyze and generate standardized AMR surveillance reports from their routine microbiology and hospital admission data files on site [13]. In 2023, on behalf of the Ministry of Public Health (MoPH) Thailand, we trained and invited all 127 public referral hospitals in Thailand to utilize the AMASS using their data from 2022 [14,15]. During the training period, we observed that clinical microbiology laboratories utilized a wide range of automated machines and LIMS. Many

faced challenges in analyzing data stored in their LIMS. A few clinical microbiology laboratories did not use any software and had no electronic data.

The goal of this study is to assess the utilization of automated systems and LIMS for blood culture specimens, and the barriers to data analysis in Thailand.

## Materials and methods

### Study setting

In 2022, Thailand had a population of 66.1 million, consisted of 77 provinces, and covered 513,120 km². The health systems in each province were integrated into 12 groups of provinces, known as health regions, plus the capital Bangkok as health region 13, using the concept of decentralization [16]. The Health Administration Division, MoPH supervised 127 public referral hospitals in health regions 1 to 12. These included 35 advanced-level referral hospitals (i.e., level-A, with a bed size of about 500-1,200), 55 standard-level referral hospitals (i.e., level-S, with a bed size of about 300-500) and 37 mid-level referral hospitals (i.e., level-M1, with a bed size of about 180-300) [17]. All level-A and S hospitals, and most of level-M1 hospitals were equipped with a microbiology laboratory capable of performing bacterial culture using standard methodologies for bacterial identification and susceptibility testing provided by the Department of Medical Sciences, MoPH [18].

### The questionnaire

We developed an online questionnaire, comprising questions about (a) the baseline characteristics of the hospital and microbiology laboratory (e.g., the hospital information system [HIS], microbiology LIMS, and the methods used to enter data of blood culture specimens into the LIMS and HIS), (b) blood culture (e.g., the total number of blood culture bottles processed, the methods and automated systems utilized for blood culture incubation, and the methods used to enter result data [i.e., growth and no-growth] into the LIMS and HIS), (c) bacterial identification (e.g., the methods and automated systems utilized for bacterial identification, and the methods used to enter result data [i.e., bacterial species] into the LIMS and HIS), (d) AST (e.g., the methods and automated systems utilized for AST, and the methods used to enter AST result data into the LIMS and HIS), and (e) data analysis (e.g., barriers to data analysis) in 2022. The initial questionnaire was piloted among co-authors to test the clarity of questions and answer choices. The questionnaire was revised and finalized based on the feedback. Two questions were merged into one because the respondents found that the two questions overlapped. The final questionnaire included 43 questions (S1 File).

### Study participants

We invited laboratory staff from all 127 clinical microbiology laboratories (one each from 127 public referral hospitals) to complete the online survey via three rounds of email invitation. We requested a representative from each laboratory to complete the survey. We also asked for consent to contact participants for clarification if any answers received via the survey were unclear. Sample size calculation was not performed because we surveyed all 127 public referral hospitals under the Health Administration Division of MoPH Thailand.

### Definitions

For this study, we categorized automated systems for blood culture processing into three steps: incubation, bacterial identification, and AST. The total capacity of automated blood culture incubation system was defined as the maximum number of blood culture bottles that the system can incubate and monitor simultaneously.

Manual data entry was defined as the process of manually inputting patient identifier data (including name, surname and hospital number) and testing results. Automatic data entry was defined as the absence of this manual input. For example, at the specimen reception, automatic data entry into the LIMS was defined as all of patient identifiers (including name, surname and hospital number) being transferred from HIS to LIMS, so staff did not need to manually enter these identifiers into the LIMS. For each automated system, automatic data entry was defined as all testing results being transferred from the automated system to the LIMS, so staff did not need to manually enter the results into the LIMS.

## Statistical analysis

Medians, interquartile ranges (IQR) and ranges of continuous variables were estimated. IQRs are presented in terms of 25th and 75th percentiles. We compared proportions and continuous variables between hospital levels using Fisher's exact test and Kruskal-Wallis test, respectively. A Spearman rank correlation coefficient (Spearman's rho) was used to assess correlations between two continuous variables. We used McNemar's exact test to compare proportions of barriers to data analysis prior to implementing AMASS and after utilizing AMASS. P values were given to two significant figures, and no longer than three decimal places. We used STATA (version 14.2; College Station, Texas) for the final statistical analysis.

## Ethics

Ethical permission for this study was obtained from both of the Oxford Tropical Research Ethics Committee (OxTREC 553-23) and the Institute for the Committee of the Faculty of Tropical Medicine, Mahidol University (TMEC 23-049). Individual participant consent was obtained online prior to completing the questionnaire, and the Ethics Committees approved this process.

## Results

### Baseline characteristics

From October 2023 to February 2024, 81 of 127 (63.8%) clinical microbiology laboratories completed the online survey and were included in the study. A total of 50 laboratories were contacted for clarification of answers. The participating laboratories were located in 60 of 77 provinces (78%) in Thailand. Of all public referral hospitals in Thailand, 89% of level-A hospitals (31/35), followed by 55% of level-S hospitals (30/55) and 54% of level-M1 hospitals (20/37) were included in this study. The median bed count of 81 participating hospitals was 450 (IQR 290 to 678).

In 2022, 79 of 81 participating hospitals (98%) utilized commercial HIS, while the remaining two hospitals used an inhouse database software as an HIS (Fig 1 and S1 Table). The most commonly used HIS was the HOSxP (67%, n = 54, Bangkok Medical Software, Bangkok, Thailand), followed by the HoMC (11%, n = 9, Info-d Software, Bangkok, Thailand). All commercial HIS (n = 8) utilized were from companies based in Thailand.

A total of 62 hospitals (77%) requested blood culture using both electronic-based and paper-based methods (i.e., double systems), 13 hospitals (16%) requested using only electronic-based methods (i.e., paperless) and six hospitals (7%) requested using only paper-based methods. All hospitals printed labels with patient identifiers from the HIS and applied them to the blood culture bottles before sending to the laboratories. The printed labels included a barcode in 39 hospitals (48%), and did not include a barcode in 42 hospitals (52%).

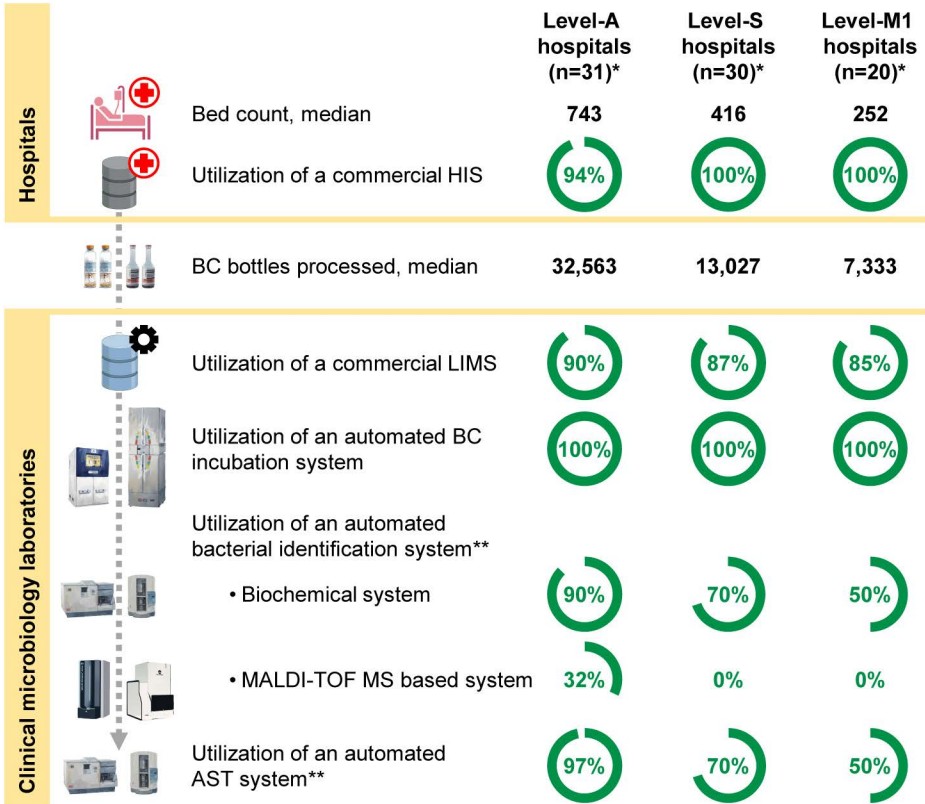

**Fig 1. Baseline characteristics.** HIS = Hospital Information System. LIMS = Laboratory Information Management System. BC = Blood culture. MALDI-TOF MS = Matrix-Assisted Laser Desorption/Ionization Time-Of-Flight Mass Spectrometry. AST = Antimicrobial Susceptibility Testing. The gray dashed line represents the flow of blood culture specimens and bacteria isolated from the blood culture specimens. *Hospital level is defined by the Ministry of Public Health (MoPH) Thailand. Level-A is Advanced-level, level-S is Standard-level, and level-M1 is Mid-level public referral hospital. **Some automated systems can perform both bacterial identification and AST. Created with BioRender.com, under a CC BY license, with permission from Biorender, original copyright 2024.

The median total number of blood culture bottles processed in 2022 was 17,351 (IQR 9,531 to 29,404). All hospitals (100%, n = 81) utilized blood culture bottles for automated systems. However, five hospitals (6%) also utilized in-house prepared blood culture bottles, which accounted for 1 to 5% of the total number of blood culture bottles processed. The total number of blood culture bottles processed was strongly associated with the hospital level (p < 0.001; S2 Table) and correlated with hospital bed count (Spearman's rho = 0.89, p < 0.001; S1 Fig).

## Clinical microbiology laboratories

The median number of full-time staff working in clinical microbiology laboratories was 4 (IQR 3 to 7; S1 Table). About one-third of the laboratories (36%, n = 29) had staff working on a 24-hour basis. About two-third of the laboratories (64%, n = 52) also processed blood culture bottles from other hospitals. The median proportion of blood culture bottles that were from other hospitals was 7% (IQR 2% to 16%, n = 52).

For daily data management of blood culture specimens, 75 laboratories (95%) used commercial LIMS, three (4%) used WHONET, three (4%) used an in-house database software, and four (5%) did not use any software. The most commonly used LIMS was the MLAB (75%, n = 61, Medical and Food Lab, Bangkok, Thailand), followed by the LAB PLUS (4%, n = 3, LAB

PLUS, Nonthaburi, Thailand) and the ALLABIS-M (4%, n = 3, ALLABIS, Thailand). All commercial LIMS (n = 8) utilized were from companies based in Thailand.

At the specimen reception (before incubating the blood culture specimens), 49 clinical microbiology laboratories (60%) entered specimen data or recorded acknowledgement of specimen reception into their commercial LIMS or other software for daily data management, 40 (49%) entered data into HIS, 17 (21%) recorded data into a paper-based lab book, and five (6%) did not enter data at this step.

About half of the laboratories (53%, n = 43) were able to automatically import patient identifier data (including names, surnames and hospital numbers) from HIS into their commercial LIMS. About one-third of the laboratories (35%, n = 28) manually entered these data into their commercial LIMS, while the remaining 10 laboratories (12%) did not have a commercial LIMS (Fig 2). The capability to automatically import data from HIS into LIMS was more common in larger hospitals (74% of level-A, 57% of level-S, and 15% of level-M1 hospitals; p < 0.001; S3 Table).

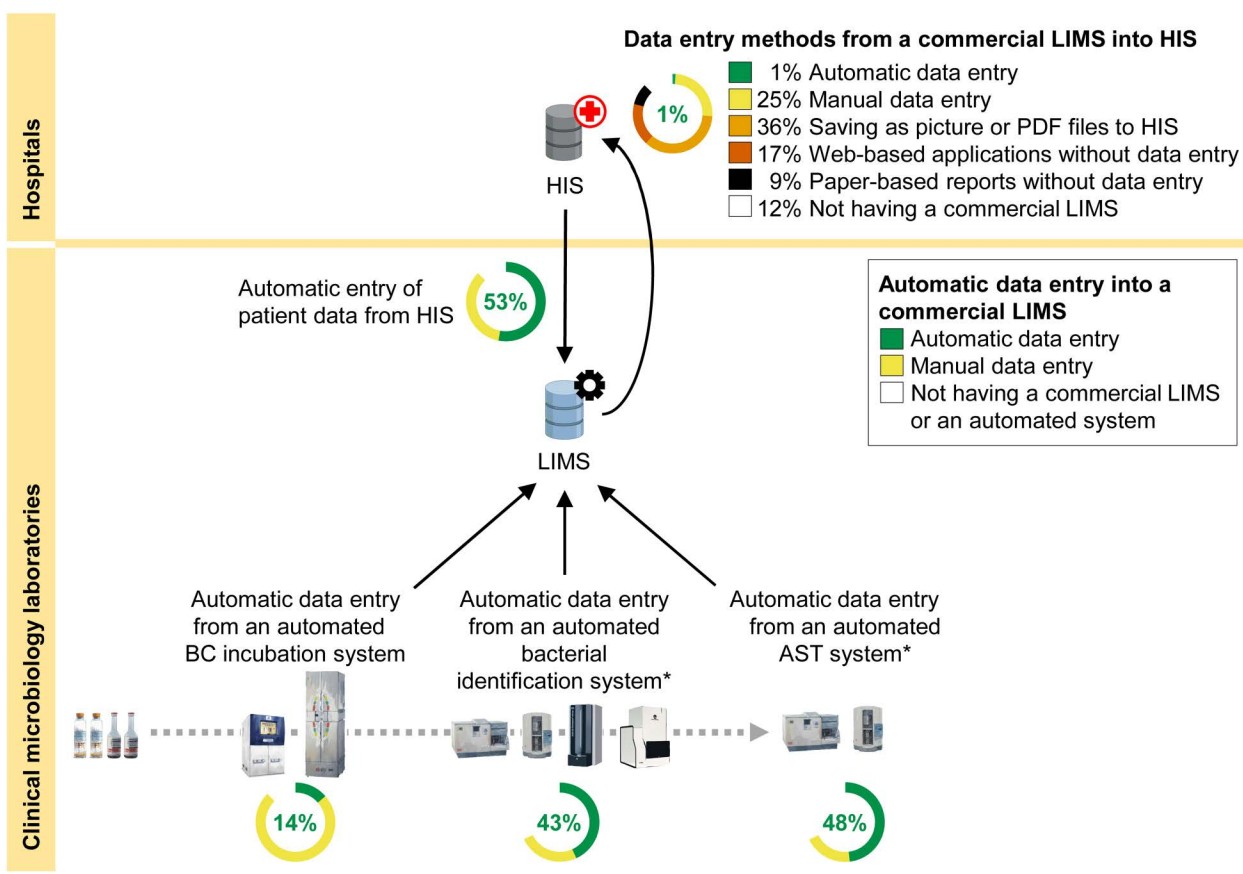

**Fig 2. Data entry methods to and from commercial LIMS.** HIS = Hospital Information System. LIMS = Laboratory Information Management System. BC = Blood culture. AST = Antimicrobial Susceptibility Testing. The black lines represent the flow of data. The gray dashed line represents the flow of blood culture specimens and bacteria isolated from the blood culture specimens. Web-based applications included both intranet and internet applications. *Some automated systems can perform both bacterial identification and AST. Created with BioRender.com, under a CC BY license, with permission from Biorender, original copyright 2022.

## Blood culture – incubation step

All clinical microbiology laboratories (100%, n = 81) had a commercial-automated blood culture incubation system (Fig 1 and S2 Table). The median capacity of automated incubation system was 400 bottles (IQR 240-600, range 120 to 1,200). The most common automated system was the Bactec system (59%, n = 48, Becton Dickinson, Sparks, MD, USA), followed by the BacT/Alert system (26%, n = 21, BioMérieux, Marcy-l'Étoile, France).

About one-seventh of the laboratories (14%, n = 12) were able to automatically import growth and no-growth results from their automated blood culture incubation systems into their commercial LIMS. Most laboratories (74%, n = 60) manually entered growth and no-growth results from their automated systems into their commercial LIMS, while the remaining 10 laboratories (12%) did not have a commercial LIMS (Fig 2). The capability to automatically import data from an automated blood culture incubation system was more common in larger hospitals (35% of level-A, 3% of level-S, and 0% of level-M1 hospitals; p = 0.004; S3 Table).

## Blood culture – bacterial identification step

Of the 81 laboratories, two (2%) outsourced the bacterial identification and AST steps to other clinical microbiology laboratories. Both hospitals were small public referral hospitals (i.e., level-M1 hospitals).

Three-quarters of the laboratories (75%, n = 61) utilized at least one automated bacterial identification system (S2 Table). The most common automated biochemical identification system utilized was the Vitek system (44%, n = 36, BioMerieux, Marcy-l'Étoile, France), followed by the Sensititre system (20%, n = 16, ThermoFisher Scientific, MA, USA). Ten advanced-level public referral hospitals (i.e., level-A hospitals) also utilized a MALDI-TOF MS-based identification system. The most common automated MALDI-TOF MS-based identification system was the Bruker MALDI-TOF MS system (6% n = 5, Becton Dickinson, Sparks, MD, USA). Utilization of an automated bacterial identification system was higher in larger hospitals (97% in level-A, 70% in level-S and 50% in level-M1, p < 0.001, S2 Table).

About two-thirds of the laboratories (67%, n = 54) routinely utilized both automated systems and conventional methods for bacterial identification, about a quarter (22%, n = 18) utilized only conventional methods, about one-tenth (9%, n = 7) utilized only automated systems, while two hospitals (2%) outsourced this step. The median proportion of bacterial isolates from blood culture specimens being identified using an automated system was higher in larger hospitals (99% in level-A, 90% in level-S and 3% in level-M1 hospitals, p = 0.008).

Less than half of clinical microbiology laboratories (43%, n = 35) were able to automatically import bacterial identification results from their automated systems into their commercial LIMS. About one-fourth of the laboratories (25%, n = 20) manually entered these data into their commercial LIMS, while the remaining 26 laboratories (32%) did not have a commercial microbiology LIMS or an automated system for bacterial identification. The capability to automatically import data from their automated bacterial identification systems was more common in larger hospitals (55% of level-A, 50% of level-S and 15% of level-M1 hospitals, p = 0.003; S3 Table).

## Blood culture – AST step

Three-quarters of the laboratories (75%, n = 61) utilized at least one automated AST system. The most common automated AST system utilized was the Vitek system (43%, n = 35, Biomeieux, Marcy-l'Étoile, France), followed by the Sensititre system (26%, n = 21, ThermoFisher Scientific, MA, USA; S2 Table). These 61 hospitals also had at least one automated system for bacterial identification.

All laboratories with an automated AST system (75%, n = 61) also routinely utilized conventional methods for AST. The median proportion of bacterial isolates from blood culture specimens being tested for antimicrobial susceptibility using an automated system was also higher in larger hospitals (95% in level-A, 50% in level-S and 3% in level-M1 hospitals, p = 0.012).

About half of clinical microbiology laboratories (48%, n = 39) were able to automatically import bacterial identification results from their automated systems into their commercial LIMS. About one-fifth of the laboratories (20%, n = 16) manually entered these data into their commercial LIMS, while the remaining 26 laboratories (32%) did not have a commercial microbiology LIMS or an automated system for AST. The capability to automatically import data from their automated AST systems was also more common in larger hospitals (p = 0.002; S3 Table).

Overall, after completing both bacterial identification and AST steps, only one laboratory (1%) was able to automatically transfer final blood culture results from their LIMS into HIS. About one-fourth of the laboratories (25%, n = 20) manually entered final blood culture results into HIS, and about one-third of the laboratories (36%, n = 29) generated a picture or PDF file from their LIMS and saved the file to the HIS. About one-seventh of the laboratories (17%, n = 14) utilized a web-based application to display results from their commercial LIMS to healthcare workers without entering or transferring data into HIS, and about one-eighth of the laboratories (12%, n = 10) utilized only paper-based reports without entering or transferring data into HIS.

## Blood culture data – analysis

All but one laboratory (99%, n = 80) stated that they generated cumulative antimicrobial susceptibility testing (cAST) reports for the year 2022. Prior to implementing AMASS, the laboratories stated that 'lack of time' (59%, n = 48), 'lack of staff with statistical skills' (48%, n = 39), 'difficulty in using an analytical software' (51%, n = 41), 'difficulty in importing data into an analytical software' (46%, n = 37) and 'difficulty in exporting data from their LIMS or other software used for daily data management (21%, n=17)' were major or very major problems for data analysis (Fig 3a).

After utilizing AMASS, the proportion of laboratories stating 'lack of time' as a major or very major problem decreased from 59% to 40% (p = 0.002, Fig 3b). Similarly, the proportion of laboratories stating 'lack of staff with skills (to use AMASS)' decreased from 48% to 32% (p = 0.035), 'difficulty in using analytical software (AMASS)' decreased from 51% to 20% (p < 0.001), and 'difficulty in importing data into analytical software (AMASS)' decreased from 46% to 26% (p = 0.004). However, the proportion of laboratories stating 'difficulty in exporting data from their LIMS or other software used for daily data management' as a major or very major problem was not different (21% vs. 17%, p = 0.58).

## Discussion

Our study shows that a variety of automated systems for blood culture and microbiology LIMS are utilized in clinical microbiology laboratories in Thailand. However, a lack of connectivity between the microbiology LIMS and each automated system, as well as between a microbiology LIMS and HIS, is common. This lack of connectivity necessitates manual data entry, therefore increasing workload and potential for errors [19]. Additionally, some hospitals utilized web-based applications without data entry into HIS, and the lack of data entry into HIS also hampers the potential utilization of microbiology data for cluster alert systems [2] and digital decision support systems [20] based on patients' electronic medical record in

**A**

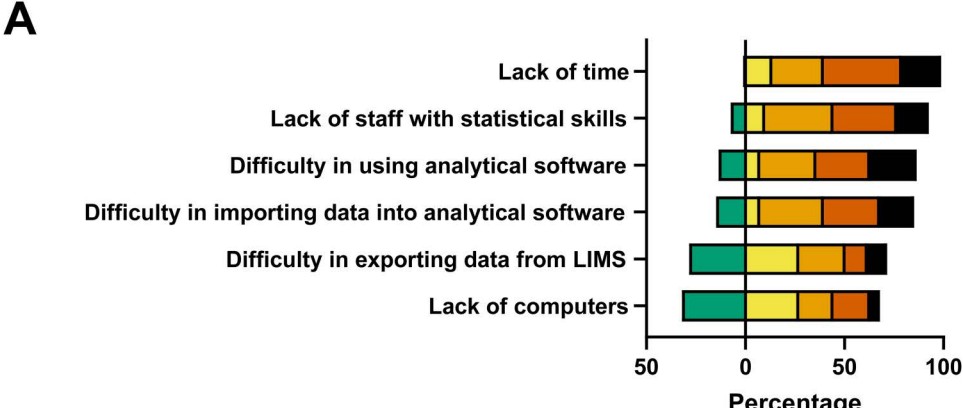

**B**

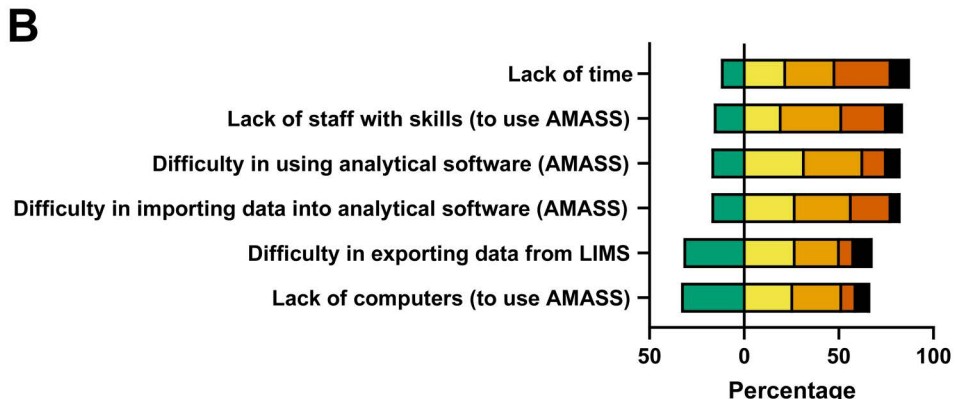

**Fig 3. Barriers to data analysis (A) prior to implementing AMASS and (B) after utilizing AMASS.** Green represents 'no problem,' yellow represents 'minor problem,' orange represents 'medium problem,' deep orange represents 'major problem,' and black represents 'very major problem'.

the HIS. In addition, many clinical microbiology laboratories have various barriers to data analysis. While the implementation of AMASS reduces these barriers to some extent, it cannot eliminate them.

The data on utilization of automated systems, capacity of automated blood culture systems, and the use of LIMS across different levels of public hospitals in Thailand could serve as one example to support the enhancement of clinical microbiology laboratories in other upper-middle-income countries. The capacity of clinical microbiology laboratories can be increased through multifaceted interventions, including advocacy, commitment of funding, enhancing education and training, care for the laboratory workforce, metrics for monitoring, etc. [21]. However, lower-middle and low-income countries may require financial and non-financial support from various international and national organizations to increase capacity on clinical microbiology laboratories [22]. A recent study in Kenya, a lower-middle-income country, also suggests that multiple issues need to be targeted simultaneously and continuously at the health system level to sustain and provide routine blood culture testing in the country [23].

The widespread adoption of automated blood culture incubation systems in all large and small public referral hospitals in Thailand is probably due to their cost-effectiveness, particularly when the number of blood culture bottles processed each day is high enough [24]. Acting

as reference centers, the advanced referral public hospitals (i.e., level-A hospitals) are increasingly equipped with a MALDI-TOF MS-based system for bacterial identification. Although the automated systems for bacterial identification and AST are increasingly utilized, utilization of both conventional methods and automated systems for these two steps is still common in other upper-middle-income and high-income countries [25,26].

Interestingly, a few small public referral hospitals utilize an automated blood culture incubation system, but outsource the bacterial identification and AST steps. This approach may be cost-effective; however, the blood culture results are reported back to the hospitals via web-based applications without data entry into the HIS and without providing a batch data file for the laboratories to conduct their own data analysis. We discussed this issue with the hospitals and the MoPH, and we recommend that future contracts with outsource laboratories include a provision to provide batch data files for the hospitals. Additionally, hospitals should explore opportunities to enhance their capacity and data utilization in the future.

Barriers to obtaining and analyzing data to generate cAST reports are consistent with those reports from other LMICs [27,28]. In Thailand, the national AMR surveillance programme (NARST) has been supporting hospitals by receiving data from each hospital, and generate national and facility-specific cAST reports annually [9,29]. Nonetheless, exporting data from LIMS and importing it into an analytical software including WHONET are also reported as barriers. A study from Uganda, a low-income country, describes the use of a Microsoft Access database to record clinical and laboratory data, the problem of connectivity, and the transition to a web-based data management system that can serve as the basis for responsive and sustainable public health surveillance [30]. These barriers are uncommon in high-income countries, where leading HIS or EHR (electronical health records) software such as EPIC [31] can cover the functionality of a microbiology LIMS and be used to produce cAST reports automatically [32]. We propose that hospitals in LMICs that have electronic data should use any appropriate and affordable (or open-access) analytical software to independently analyzed and utilize their data for immediate actions [14,15]. Technologies should also be carefully developed and implemented to reduce these barriers in LMICs [33].

Our study has multiple limitations. First, our study did not aim to develop guidance for LIMS. It is recommended to consider multiple factors, including relevance, compliance, compatibility, cost, speed, connectivity, adaptability and reliability [28,34]. Second, our study did not evaluate a general laboratory information management system (LIMS), commonly used in clinical biochemical laboratories [35]. Third, it is unknown whether non-responding hospitals utilize automated systems and LIMS similar to the responding hospitals.

In conclusion, various automated systems for blood culture and LIMS are utilized in Thailand. However, barriers to data management and analysis are common. These challenges are likely present in other upper-middle-income countries. To improve efficiency, quality and benefits of clinical microbiology laboratories, we propose that guidance and technical support for automated systems, LIMS, and data analysis are required.

## Supporting information

**S1 Table. Baseline characteristics of 81 public referral hospitals in Thailand, 2022.**
(DOCX)

**S2 Table. Utilization of automated systems for blood culture.**
(DOCX)

**S3 Table. Data entry methods to and from commercial LIMS.**
(DOCX)

**S1 Fig. Correlation between the total number of blood culture bottles and hospital bed count.**
(TIF)

**S1 File. Questionnaire.**
(PDF)

## Acknowledgments

We gratefully acknowledge the survey participants for their contribution to the study.

## Author contributions

**Conceptualization:** Preeyarach Klaytong, Paul Turner, Direk Limmathurotsakul.

**Data curation:** Preeyarach Klaytong, Direk Limmathurotsakul.

**Formal analysis:** Preeyarach Klaytong, Direk Limmathurotsakul.

**Funding acquisition:** Direk Limmathurotsakul.

**Investigation:** Preeyarach Klaytong, Panida Chamawan, Voranadda Srisuphan, Krittiya Tuamsuwan, Phairam Boonyarit, Adisak Sangchankoom, Archawin Rojanawiwat, Aekkawat Unahalekhaka, Pimrata Leethongdee, Orapan Sripichai, Vanaporn Wuthiekanun, Direk Limmathurotsakul.

**Methodology:** Preeyarach Klaytong, Panida Chamawan, Voranadda Srisuphan, Krittiya Tuamsuwan, Phairam Boonyarit, Adisak Sangchankoom, Archawin Rojanawiwat, Aekkawat Unahalekhaka, Pimrata Leethongdee, Orapan Sripichai, Vanaporn Wuthiekanun, Paul Turner, Direk Limmathurotsakul.

**Resources:** Kulsumpun Krobanan, Direk Limmathurotsakul.

**Software:** Kulsumpun Krobanan.

**Supervision:** Direk Limmathurotsakul.

**Validation:** Preeyarach Klaytong, Adisak Sangchankoom.

**Writing – original draft:** Preeyarach Klaytong, Direk Limmathurotsakul.

**Writing – review & editing:** Preeyarach Klaytong, Panida Chamawan, Voranadda Srisuphan, Krittiya Tuamsuwan, Phairam Boonyarit, Adisak Sangchankoom, Aekkawat Unahalekhaka, Kulsumpun Krobanan, Pimrata Leethongdee, Orapan Sripichai, Vanaporn Wuthiekanun, Paul Turner, Direk Limmathurotsakul.

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
