## [Decision Letter · Decision Letter 0]

14 Jan 2025

Dear Dr. Limmathurotsakul,

Thank you for submitting your manuscript to PLOS ONE. After careful consideration, we feel that it has merit but does not fully meet PLOS ONE’s publication criteria as it currently stands. Therefore, we invite you to submit a revised version of the manuscript that addresses the points raised during the review process.

1. The authors made it clear Thailand is not LMIC. However, Abstract's Introduction states: little is known about how data are generated, managed and analyzed in those laboratories in low and middle-income countries (LMICs). This knowledge gap cannot be filled by this study. So the authors would better not ask this question as it will cause confusion. Please correct this across the manuscript. For example, line 343-344: "Our findings can support the global initiative to strengthen microbiology laboratory capacity in LMICs". 

2. Line 359-361 in Discussion: "Although the automated systems for bacterial identification and AST are increasingly utilized, utilization of both conventional methods and automated systems for these two steps is still common." There are no references for this statement. More references are needed, with this reference (PMID: 34142857) as an example (citing is optional).

We look forward to receiving your revised manuscript.

Kind regards,

Benjamin M. Liu, MBBS, PhD, D(ABMM), MB(ASCP)

Academic Editor

PLOS ONE

3. Thank you for stating the following financial disclosure:  [The study was supported by Ministry of Public Health Thailand and Wellcome Trust of Great Britain (224681/Z/21/Z). For the purpose of Open Access, the author has applied a CC BY public copyright license to any Author Accepted Manuscript version arising from this submission.]. 

Additional Editor Comments (if provided):

Reviewers' comments:

Reviewer's Responses to Questions

Comments to the Author

1. Is the manuscript technically sound, and do the data support the conclusions?

Reviewer #1: Yes

Reviewer #2: Yes

2. Has the statistical analysis been performed appropriately and rigorously?

Reviewer #1: Yes

Reviewer #2: Yes

3. Have the authors made all data underlying the findings in their manuscript fully available?

Reviewer #1: Yes

Reviewer #2: Yes

4. Is the manuscript presented in an intelligible fashion and written in standard English?

Reviewer #1: No

Reviewer #2: Yes

Reviewer #1: Dear Authors

Though your manuscript and work is good but i suggest some changes.

1.Your title is a little bit confusing and doesn't provide the goal of the study.

2. Please make some changes in key words.

3. Delete the repeated citation in line 65 and make it whole with reference number 3.

4. The language quality and level must get enhance by a native speaker.

5. Some of introduction sentences is not related to the subject. provide better introduction.

6. Where is the conclusion part?

7. In Study participants section you have mentioned inviting laboratory staff from clinical microbiology laboratories, was it random or you did it based on something?

8. Discussion section is more like introduction part, you should mention about other related studies ( if there are any) and compare your research.

9. Explain more about the goal of the study.

Kind regards

Reviewer #2: This manuscript although seems very basic to the standrads of international publications but has weight when it comes to understanding the present requirements in clinical set up. Health care system is rapidly improving and also evolving with the big data. One of the main concern is on how to maintain and handle big data for patient benefit.

In this scenario, this manuscript presents valid observation based on multi-location data to guide towards need of a better network based data handling. I am happy to see these kind of studies which involves enormous amount of time and compilation which might yield a small conclusion but in a positive manner to focus on priorities and necessities. I recommend to accept this MS which can be a valuable resource for patient data handling and management.

**Do you want your identity to be public for this peer review?** For information about this choice, including consent withdrawal, please see our Privacy Policy

Reviewer #1: No

Reviewer #2: No

---

## [Author Response · Author response to Decision Letter 1]

10 Feb 2025

Re: PLOS ONE Decision: Revision required [PONE-D-24-59742] - [EMID:5b975ce039a7bcd5]

Dear Editor,

Thank you for your e-mail concerning the above manuscript. We are grateful for the helpful comments made by the reviewers, and have addressed them as detailed below.

Editor's comments:

1. The authors made it clear Thailand is not LMIC. However, Abstract's Introduction states: little is known about how data are generated, managed and analyzed in those laboratories in low and middle-income countries (LMICs). This knowledge gap cannot be filled by this study. So the authors would better not ask this question as it will cause confusion. Please correct this across the manuscript. For example, line 343-344: "Our findings can support the global initiative to strengthen microbiology laboratory capacity in LMICs".

Answer. The sentence in the abstract has been revised as follows, “Here, we aim to understand the systems involved in generating, managing and analyzing blood culture data in these laboratories in an upper-middle income country.” The sentence in the discussion has been removed and the discussion has been revised for clear difference between upper-middle-income and lower-middle-income and low-income countries as follows, “The data on utilization of automated systems, capacity of automated blood culture systems, and the use of LIMS across different levels of public hospitals in Thailand can serve as one example to support the rational installation and enhancement of clinical microbiology laboratories in other upper-middle-income countries. The capacity of clinical microbiology laboratories can be increased through multifaceted interventions, including advocacy, commitment of funding, enhancing education and training, care for the laboratory workforce, metrics for monitoring, etc. [21]. Many lower-middle and low-income countries may require financial and non-financial support from various international and national organizations to increase capacity on clinical microbiology laboratories [22]. ”

2. Line 359-361 in Discussion: "Although the automated systems for bacterial identification and AST are increasingly utilized, utilization of both conventional methods and automated systems for these two steps is still common." There are no references for this statement. More references are needed, with this reference (PMID: 34142857) as an example (citing is optional).

Answer. The new reference (PMID: 34142857) and the reference no. 21 (PMID: 31275940) have been added for the statement.

Answer. Revised as suggested.

Answer. The ethics statement has been revised as follows, “Ethical permission for this study was obtained from both of the Oxford Tropical Research Ethics Committee (OxTREC 553-23) and the Institute for the Committee of the Faculty of Tropical Medicine, Mahidol University (TMEC 23-049). Individual participant consent was obtained online prior to completing the questionnaire, and the Ethics Committee approved this process.”

3. Thank you for stating the following financial disclosure: [The study was supported by Ministry of Public Health Thailand and Wellcome Trust of Great Britain (224681/Z/21/Z). For the purpose of Open Access, the author has applied a CC BY public copyright license to any Author Accepted Manuscript version arising from this submission.].

Answer. The following sentence has been added for clarity, “The funders had no role in study design, data collection and analysis, decision to publish, or preparation of the manuscript.”

4. When completing the data availability statement of the submission form, you indicated that you will make your data available on acceptance. We strongly recommend all authors decide on a data sharing plan before acceptance, as the process can be lengthy and hold up publication timelines.

Answer. The data has been made open-access, and the sentence has been revised as follows, “The dataset generated for this study is available at https://doi.org/10.6084/m9.figshare.28028285.”

Answer. Figure 1 has been removed.

Reviewer #1: Dear Authors

Though your manuscript and work is good but i suggest some changes.

1.Your title is a little bit confusing and doesn't provide the goal of the study.

Answer. The title has been revised to include the goal of the study as follows, “Assessment of Utilization of Automated Systems and Laboratory Information Management Systems in Clinical Microbiology Laboratories in Thailand”

2. Please make some changes in key words.

Answer. The keyword “low and middle-income countries” has been changed to “upper-middle-income countries” for clarity.

3. Delete the repeated citation in line 65 and make it whole with reference number 3.

Answer. The repeated citation in line 65 has been deleted as suggested.

4. The language quality and level must get enhance by a native speaker.

Answer. The manuscript's language has been revised by co-author Paul Turner, a native English speaker.

5. Some of introduction sentences is not related to the subject. provide better introduction.

Answer. We would like to maintain the sentences in the introduction because they provide background on why we conducted the study and why we assessed the barriers following the implementation of AMASS.

6. Where is the conclusion part?

Answer. A conclusion paragraph has been added as follows, “In conclusion, various automated systems for blood culture and LIMS are utilized in Thailand. However, barriers to data management and analysis are common. These challenges are likely present in other upper-middle-income countries. To improve efficiency, quality and benefits of clinical microbiology laboratories, we propose that guidance and technical support for automated systems, LIMS, and data analysis are required.”

7. In Study participants section you have mentioned inviting laboratory staff from clinical microbiology laboratories, was it random or you did it based on something?

Answer. We requested all clinical microbiology laboratories in all 127 public referral hospitals to participate the study. No randomization was performed. The sentence has been revised for clarity, “We invited laboratory staff from all 127 clinical microbiology laboratories (one each from 127 public referral hospitals) to complete the online survey via three rounds of email invitation.”

8. Discussion section is more like introduction part, you should mention about other related studies (if there are any) and compare your research.

Answer. Additional references and sentences have been added as follows, “A recent study in Kenya, a lower-middle-income country, also suggests that multiple issues need to be targeted simultaneously and continuously at the health system level to sustain and provide routine blood culture testing in the country [23].”, “Although the automated systems for bacterial identification and AST are increasingly utilized, utilization of both conventional methods and automated systems for these two steps is still common in other upper-middle-income and high-income countries [25, 26].” and “A study from Uganda, a low-income country, describes the use of a Microsoft Access database to record clinical and laboratory data, the problem of connectivity, and the transition to a web-based data management system that can serve as the basis for responsive and sustainable public health surveillance [30].”

9. Explain more about the goal of the study.

Answer. The sentence has been revised for clarity as follows, “The goal of this study is to assess the utilization of automated systems and LIMS for blood culture specimens, and the barriers to data analysis in Thailand.”

Kind regards

Reviewer #2: This manuscript although seems very basic to the standrads of international publications but has weight when it comes to understanding the present requirements in clinical set up. Health care system is rapidly improving and also evolving with the big data. One of the main concern is on how to maintain and handle big data for patient benefit.

In this scenario, this manuscript presents valid observation based on multi-location data to guide towards need of a better network based data handling. I am happy to see these kind of studies which involves enormous amount of time and compilation which might yield a small conclusion but in a positive manner to focus on priorities and necessities. I recommend to accept this MS which can be a valuable resource for patient data handling and management.

Answer. We are grateful for the comment.

All contributing authors have reviewed and concurred with the revised manuscript.

Yours,

Direk Limmathurotsakul

---

## [Editor Report · Decision Letter 1]

13 Feb 2025

Assessment of utilization of automated systems and laboratory information management systems in clinical microbiology laboratories in Thailand

PONE-D-24-59742R1

Dear Dr. Limmathurotsakul,

We’re pleased to inform you that your manuscript has been judged scientifically suitable for publication and will be formally accepted for publication once it meets all outstanding technical requirements.

Kind regards,

Benjamin M. Liu, MBBS, PhD, D(ABMM), MB(ASCP)

Academic Editor

PLOS ONE
---

## [Editor Report · Acceptance letter]

PONE-D-24-59742R1

PLOS ONE

Dear Dr. Limmathurotsakul,

I'm pleased to inform you that your manuscript has been deemed suitable for publication in PLOS ONE. Congratulations! Your manuscript is now being handed over to our production team.

Kind regards,

on behalf of

Dr. Benjamin M. Liu

Academic Editor

PLOS ONE